# Challenging the Binary Classification of Endometrioid Endometrial Adenocarcinoma: The Evaluation of Grade 2 as an Independent Entity Based on Prognostic Characteristics and Recurrence Patterns [note 1]

**DOI:** 10.3390/cancers17010127

**Published:** 2025-01-03

**Authors:** Andreas Zouridis, Ammara Kashif, Ahmed Darwish, Christina Pappa, Federico Ferrari, Sarah Louise Smyth, Negin Sadeghi, Alisha Sattar, Stephen Damato, Mostafa Abdalla, Sean Kehoe, Susan Addley, Hooman Soleymani Majd

**Affiliations:** 1Oxford University Hospitals NHS Foundation Trust, Oxford OX3 7LE, UKahmed.darwish@ouh.nhs.uk (A.D.);; 2Department of Clinical and Experimental Sciences, University of Brescia, 25136 Brescia, Italy; federico.ferrari@unibs.it; 3Guy’s and St Thomas’ NHS Foundation Trust, London SE1 9RT, UK; 4University Hospitals of Derby and Burton NHS Foundation Trust, Derby DE22 3NE, UK

**Keywords:** G2, grade 2, low grade, moderately differentiated, endometrioid endometrial cancer

## Abstract

The traditional three-tier grading system for endometrioid endometrial cancer has been replaced in research—and recently in clinical practise—by a binary system that combines grade 1 and grade 2 tumours under the low-grade umbrella. This simplification attempts to overcome diagnostic clinicopathological challenges but overlooks the unique nature of grade 2 endometrioid endometrial tumours. The aim of the present study is to investigate the unique prognostic characteristics and recurrence patterns of grade 2 endometrioid endometrial cancers and elucidate where the prognosis of grade 2 tumours stands in the spectrum between grade 1 and grade 3 cancers.

## 1. Introduction

In 2020, 417,000 women were diagnosed and 97,000 died of uterine corpus cancer, the sixth most common cancer in women [1]. Almost 80% of endometrial cancers are endometrioid endometrial carcinomas (EECs), which are traditionally characterised as grade 1 (G1), or well differentiated; grade 2 (G2), or moderately differentiated; and grade 3 (G3), or poorly differentiated, according to architectural features and the extent of nuclear atypia [2].

Current grading criteria, according to the WHO Classification of Tumours, are primarily based on architectural features but also consider cytologic atypia [3]. Histological grade has always been one of the widely recognised prognostic factors for EEC [4]. However, many clinicians combine G1 and G2 tumours to a single low-grade category to increase inter- and intra-observer agreement and the reproducibility of diagnosis [5]. This binary tumour grading system has been adopted by FIGO [6] and ESMO/ESTRO/ESP [7].

The switch from the traditional three-tier to the binary grading system was not imposed by the prognostic similarities between G1 and G2 but was a result of the diagnostic challenge of distinguishing between G1 and G2. The wide use of the binary grading system is the main reason for the paucity of evidence about the prognosis of G2 EEC. The aim of the present study is to investigate the prognostic characteristics and recurrence patterns of G2 ECC and elucidate where the prognosis of G2 tumours stands in the spectrum between G1 and G3 EEC.

## 2. Materials and Methods

Our retrospective study included all patients with EEC treated at Churchill Cancer Centre and John Radcliffe Women’s Centre, Oxford, between March 2010 and January 2020. The following exclusion criteria were implemented: a non-endometrioid histology, the presence of synchronous malignancy, primary non-surgical management, and suboptimal follow up (patient with less than 2 years follow up) [8].

Demographic, treatment, and survival data were retrieved from electronic and hard copy records of patients for a service evaluation project on endometrial cancer. The relevant protocol was registered (registration number 5832) [9] and approved by the Institutional Review Board of Oxford University Hospitals NSH Trust (ID: 5832, Ref:06/01/2020-SUWON-Soleymani-2). The data collected were anonymized and all patients had signed a relevant consent form before surgery. The study design complies with the Helsinki Declaration, the Committee on Publication Ethics guidelines [10] and the Reporting of studies Conducted using Observational Routinely collected health Data (RECORD) Statement validated by the Enhancing the Quality and Transparency of Health Research Network [11,12].

The initial treatment of all the patients of the study was surgery (laparoscopic or open hysterectomy and bilateral salpingo-oophorectomy). Pelvic lymphadenectomy, omental biopsy and adjuvant treatment was offered to all except from low-risk patients [13]. Follow up was conducted with clinical examination at three-month intervals for the first year after treatment, four-month intervals for the second year and annually thereafter [13]. Grade was confirmed in the final histology as follows: G1: less than 5% of a nonsquamous or nonmorular solid growth pattern; G2: 6–50% of a nonsquamous or nonmorular solid growth pattern; and G3: greater than 50% of a nonsquamous or nonmorular solid growth pattern [14]. Excessive nuclear atypia raised the grade from G1 and G2 to G2 and G3, respectively [14].

Patients’ comorbidities were summarised using the Age-Adjusted Charlson Comorbidity score (AAC score) [15] and patients were divided according to their fitness in three groups (0–1, 2–3 and >3). We have also retrieved data regarding FIGO 2009 Stage [16], the depth of myometrial invasion, cervical stromal involvement, serosal breaching, adnexal, parametrial and pelvic lymph node involvement, the presence of distant metastases, and the presence of lymphovascular invasion (LVSI). All cases were classified according to the ESGO-ESTRO-ESP risk stratification model [7]. Finally, we collected data regarding surgical approach, pelvic lymphadenectomy, and the administration of adjuvant treatment.

Statistical analysis was performed using IBM^©^SPSS Statistics 22.0. and statistical significance was considered for *p* < 0.05. Kaplan–Meier Curves and a log-rank test were used to calculate and compare survival rates. The contribution of potential risk factors to relapse and mortality for G2 endometrioid endometrial cancer was assessed using univariate and multivariate Cox proportional hazards analysis.

## 3. Results

Out of the 523 patients with endometrioid endometrial cancer of our cohort, 238 (45.5%) were G1, 189 (36.1%) G2 and 96 (18.4%) were high grade.

Patients with G2 EEC had a mean age of 67.29 years (range 26–91 years) and mean BMI of 32.83kg/m^2^ (range 16–70.9). Of those patients, 92.5% had a laparoscopic surgical staging, but only 35.4% had pelvic lymphadenectomy/sampling, with an average of 14.25 removed nodes (range 1–25). In total, 85.8% of patients with G2 EEC received the indicated (where indicated) adjuvant treatment, whereas 14.2% of patients did not, either because they were not fit, or because they declined. Most of the patients (83.1%) with G2 EEC were diagnosed with stage I or stage II disease (early stage) and in almost two thirds (66.1%) there was lymphovascular invasion present on final histology. Demographic data, treatment details and the clinicopathological characteristics of G2 EEC are summarised in Table 1.

Overall, 5-year and cancer-specific survival for G2 EEC was 88% (mean 131.54 months; 95% CI 125.37–137.71) and 93.3% (mean 139.37 months; 95% CI 134.25–144.49) respectively. Recurrence rate was 8.5% and 5-year disease-free survival was 91.6% (mean 139.95 months; 95% CI 134.76–145.15).

Patients with G2 EEC had comparable 5-year disease-free survival rates (91.6% vs. 93.2%, *p* = 0.44), but worse 5-year cancer-specific survival (93.3% vs. 98.5%, *p* < 0.01) compared to patients with G1 EEC. On the other hand, G2 EEC has a favourable prognosis compared to G3 EEC, both in terms of disease-free survival (91.6 vs. 83.8%, *p* = 0.04) and cancer-specific survival (93.3% vs. 83.6%, *p* < 0.01) (Figure 1). Multivariable Cox analysis showed that only stage is an independent risk factor for recurrence and that both grade and stage are independent risk factors for cancer-specific mortality in endometrioid endometrial cancer after adjusting for LVSI stage and grade.

Cox proportional hazard ratio analysis including multiple variables (AACCS, pelvic lymphadenectomy, adjuvant treatment, depth of myometrial invasion, cervical stromal invasion, adnexal involvement, serosal breach, parametrial invasion, pelvic lymph node involvement, distant metastases, and LVSI) showed that none of the above are independent risk factors for recurrence for G2 EEC. On the other hand, cervical stromal involvement (OR = 10.10, *p* = 0.04), parametrial involvement (OR = 40.88, *p* < 0.01) and distant metastatic disease (OR = 547.44, *p* < 0.01) are all independent risk factors for cancer-specific mortality in moderately differentiated endometrioid endometrial cancers.

In total, 43.8% of recurrences occurred within two years and 68.8% within three years of initial treatment. One recurrence was diagnosed after discharge from the 5-year regular follow up (6.3%). The median disease-free survival among patients with recurrence was 27 months and the median survival after recurrence 63 months.

Disease recurrence was diagnosed in three asymptomatic patients; in two of them during a routine follow up examination (12.5% of recurrences) and in one as an incidental finding on a CT scan that was carried out for other indications. In total, 13 patients (81.3%) presented with symptoms before the diagnosis of recurrence, with the most common symptom vaginal bleeding and discharge, seen in 6 cases (37.5% of recurrences). Single-site recurrence was diagnosed in 5 patients (31.3% of recurrences) and single vaginal vault recurrence in three cases (18.8% of recurrences), whereas more than half the recurrent cases (56.3%) had extra-pelvic disease at initial presentation. Recurrence characteristics are summarised in Table 2.

## 4. Discussion

Our study showed a clear prognostic difference between G1, G2 and G3 endometrioid endometrial cancer, with G2 between G1 and G3 tumours in terms of 5-year cancer-specific survival (G1—98.5%, G2—93.3%, G3—83.6%; *p* < 0.05). Multivariable Cox analysis confirmed that the above difference is independent of LVSI and stage. Our findings are similar to another study which analysed 800 patients with endometrioid endometrial cancer and concluded that 5-year disease-specific survival rates for G1, G2 and G3 EEC were 97%, 94% and 76%, respectively (*p* < 0.001) [17]. A recent large cohort of 1630 patients with endometrioid endometrial cancer from Sweden confirmed the significant prognostic differences among low-grade endometrial cancers in terms of 5-year overall [94.5% (95% CI: 92.6–96.3) for G1 EEC vs. 84.9% (95% CI: 82.3–87.6) for G2 EEC] and net survival [104.4% (95% CI: 102.1–106.7) for G1EC and 95.9% (95% CI: 92.7–99.3) for G2EC]. In the same study, the recurrence rate is 3.8%, 11.3% and 12.8% for G1, G2 and G3 EEC (*p* < 0.001), respectively [18].

On the other hand, other studies failed to demonstrate any prognostic difference between G1 EEC and G2 EEC. In a cohort of 253 patients with endometrial cancer from the Netherlands, there was no difference in 5-year cancer-specific survival between G1 and G2 (92 vs. 95%), whereas a high grade was found to be a significant adverse prognostic factor (*p* < 0.001) [19]. However, in that study, the sample was small and possibly insufficient to detect subtle outcome differences. Investigators of another cohort of 776 patients with endometrioid endometrial cancer also failed to find any statistically significant differences between G1 and G2 EEC on survival indicators, despite the significant variations in major pathological parameters between the groups, but cancer-specific survival was not assessed [20].

A binary staging system is more reproducible, with less inter-observer variability compared to the ternary FIGO grading system [5,17,19]. Although combining G1 and G2 in one category overcomes the diagnostic challenges, our data suggest that the above practise compromises the prognostication and increases the heterogeneity within the low-grade group. The global implementation of molecular profiling is expected to overcome the limitations of grading and identify distinct prognostic subgroups within low-grade tumours [21,22]. However, some studies challenge the routine molecular profiling in patients with low-grade EEC, and stress the importance of accurate tumour grading and selective profiling for these patients [23].

To our knowledge, our study is the first to focus on the prognostic features of G2 EEC. We did not find any independent risk factor for recurrence, possibly because of our small sample and the low recurrence rate. On the other hand, our data suggest that cervical involvement, parametrial involvement and distant metastatic disease are all independent risk factors for cancer-related mortality in G2 EEC. Similarly, cervical stromal invasion is an independent pathological risk factor for cancer-specific mortality in high-grade endometrioid endometrial cancer [9].

The significance of cervical stromal and parametrial involvement in survival for G2 endometrial cancer highlights the importance of accurate pre-operative staging and proper surgical management. Although three-dimensional transvaginal ultrasound (3D-TVUS) and magnetic resonance imaging (MRI) are equally effective for the evaluation of cervical stromal invasion in patients with endometrial cancer [24], MRI remains the gold standard for the evaluation of both cervical and parametrial involvement [25]. Accurate pre-operative staging allows for optimal surgical planning. Radical hysterectomy does not improve outcome in patients with cervical involvement [26], but it can be considered in order to achieve complete surgical resection in patients with cervical or parametrial involvement [14]. Finally, although laparoscopic staging appears safe even in cases with cervical involvement [27,28], it is very important to take all the necessary measures in order to minimise uterine manipulation [29].

Almost half of the recurrences (43.8%) in our cohort of G2 EEC were diagnosed within two years and more than two thirds (68.8%) were within three years of treatment. Those figures are in accordance with the literature [30] and justify the need for increased vigilance for three years.

Our data suggest that for G2 EEC, the recurrence risk is relatively low for all risk groups (3.9%, 9.1%, 12.5% and 12.5% for the Low, Intermediate, High-Intermediate and High risk group, respectively). In addition, only 16.7% of recurrences among Low and Intermediate risk cases and only 10% of recurrences among High-Intermediate and High risk patients were diagnosed during examination at follow up in asymptomatic patients. Hence, clinical examination is of limited value in the early diagnosis of recurrence of G2 EEC before clinical manifestation. Moreover, 81.3% of all G2 EEC recurrences presented with symptoms and there is evidence that in symptomatic patients, diagnosis might be delayed when a routine follow up appointment is scheduled [31].

The British Society of Gynaecological Oncology (BSGE) suggests patient-initiated follow up (PIFU) for Low and Intermediate risk cancers and clinic or telephone follow up for at least 2 (and up to 5) years for High-Intermediate and High risk cases. However, the abovementioned findings suggest that PIFU can be safely expanded to all G2 endometrial cancers. Both women treated for endometrial cancer and health care providers are mostly supportive of PIFU [32], whereas traditional follow up strategies do not meet cancer patients’ needs [33]. Finally, PIFU is expected to reduce waiting times and waiting lists due to a net reduction in follow up appointments (21,653 appointments for endometrial cancer in 2022–2023 in the UK) [34].

## 5. Conclusions

In conclusion, our study suggests that G2 EEC is a distinct entity, and thus the grading system should continue to differentiate G1 EEC and G2 EEC for better prognosis interpretation. Cervical and parametrial involvement are independent risk factors for cancer-specific mortality for G2 EEC. Finally, the recurrence pattern suggests that PIFU is a cost-effective alternative for the follow up of patients with G2 EEC.

## Figures and Tables

**Figure 1 cancers-17-00127-f001:**
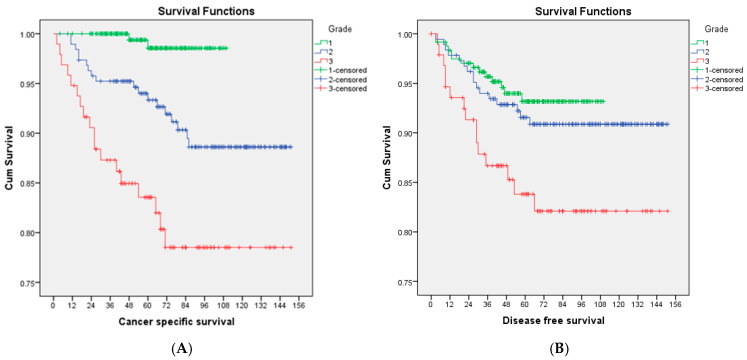
(**A**) Cancer-specific survival and (**B**) Disease-free survival of G1 EEC (green), G2 EEC (blue) and G3 EEC (red) in months.

**Table 1 cancers-17-00127-t001:** Demographic data, treatment details and clinicopathological characteristics of patients with G2 EEC.

	N (%)	Recurrence (% of Each Subgroup)	Cancer-Specific Mortality (% of Each Subgroup)
**Age**			
<65	73 (38.6)	6 (8.2)	7 (9.6)
≥65	116 (61.4)	10 (8.6)	11 (9.5)
**AACCS**			
0–1	43 (22.8)	4 (9.3)	6 (14)
2–3	93 (49.2)	2 (2.2)	3 (3.2)
>3	53 (28)	10 (18.9)	9 (17)
**Surgical approach**			
Laparoscopy	147 (92.5)	12 (8.2)	14 (9.5)
Laparotomy	12 (7.5)	2 (16.7)	2 (16.7)
**Pelvic lymph node dissection**			
No	122 (64.6)	14 (11.5)	15 (12.3)
Yes	67 (35.4)	2 (3)	3 (4.5)
**Administration of indicated adjuvant treatment**			
No	23 (14.2)	4 (17.4)	4 (14.4)
Yes	139 (85.8)	11 (7.9)	13 (9.4)
**FIGO Stage**			
IA	91 (48.1)	3 (3.3)	2 (2.2)
ΙΒ	54 (28.6)	9 (16.7)	7 (13)
ΙΙ	12 (6.3)	0 (0)	0 (0)
ΙΙΙA	12 (6.3)	2 (16.7)	2 (16.7)
ΙΙΙΒ	14 (7.4)	2 (14.3)	4 (28.6)
ΙΙΙC1	3 (1.6)	0 (0)	0 (0)
IIIC2	1 (0.5)	0 (0)	1 (100)
IVB	2 (1.1)	0 (0)	2 (100)
**Stage category**			
Early (I–II)	157 (83.1)	12 (7.6)	9 (5.7)
Advanced (III–IV)	32 (16.9)	4 (12.5)	9 (28.1)
**Myometrial invasion**			
<50%	104 (55)	5 (4.8)	6 (5.8)
≥50%	85 (45)	11 (12.9)	12 (14.1)
**Cervical stroma involvement**			
No	159 (84.1)	15 (9.4)	14 (8.8)
Yes	30 (15.9)	1 (3.3)	4 (13.3)
**Adnexal involvement**			
No	182 (96.3)	15 (8.2)	17 (9.3)
Yes	7 (3.7)	1 (14.3)	1 (14.3)
**Serosal breach**			
No	180 (95.2)	15 (8.3)	16 (8.9)
Yes	9 (4.8)	1 (11.1)	2 (22.2)
**Parametrial involvement**			
No	174 (92.1)	14 (8)	13 (7.5)
Yes	15 (7.9)	2 (13.3)	5 (33.3)
**Pelvic lymph node involvement**			
No	185 (97.9)	16 (8.6)	17 (9.2)
Yes	4 (2.1)	0	1 (25)
**Distant metastases**			
No	187 (98.9)	16 (8.6)	16 (8.6)
Yes	2 (1.1)	0	2 (100)
**LVSI**			
No	125 (66.1)	7 (5.6)	8 (6.4)
Yes	64 (33.9)	9 (14.1)	10 (15.6)
**ESGO-ESTRO-ESP Risk stratification**			
Low	76 (40.2)	3 (3.9)	2 (2.6)
Intermediate	33 (17.5)	3 (9.1)	3 (9.1)
High-intermediate	48 (25.4)	6 (12.5)	4 (8.3)
High	32 (16.9)	4 (12.5)	9 (28.1)

**Table 2 cancers-17-00127-t002:** Cases of recurrence of G2 EEC and their characteristics.

Stage	LVSI	Symptoms	Site	Treatment	Disease-Free Survival (Months)	Survival After Recurrence (Months)
IA	Absent	Vaginal bleeding	Vaginal vault + pelvic LN	BSC	29	14
IA	Absent	Persistent cough	Lungs	HT	42	33
IA	Absent	1st—asymptomatic (found on examination) 2nd—shortness of breath	1st—inguinal LN 2nd—ediastinal LN	1st SE 2nd BSC	57	29
IB	Present	Abdominal pain	Lungs + LN (supraclavicular, para-aortic, pelvic)	BSC	8	3
IB	Present	Diabetic complications + diarrhoea	Paracardiac LN + peritoneum + omentum + sigmoid	HT	11	10
IB	Absent	Persistent cough	lungs + mediastinal LN + adrenal	BSC	21	3
IB	Absent	Persistent discharge	Vaginal vault	Chemotherapy	3	24
IB	Present	Vaginal bleeding	Vaginal vault + pelvic LN (recurrence as G3 ECC)	Chemotherapy	23	43
IB	Absent	Fatigue and reduced appetite	Peritoneum—malignant ascites + LN (retrocaval, paracardiac)	BSC	63	8
IB	Present	1st—vaginal bleeding 2nd—shortness of breath	1st—vaginal vault 2nd—lungs	1st SE 2nd CT	27	58
IB	Present	Asymptomatic (incidental finding on CT scan for other reason)	Peritoneal nodularity and ascites (small volume disease) + umbilical nodule	RT + HT	31	78
IB	Present	Vaginal bleeding	Vaginal vault + pelvic tumour	SE + HT	27	107
IIIA	Present	Diarrhoea	Pelvic + paraaortic LN	RT	18	42
IIIA	Absent	Pelvic pain	Lungs + bones	RT + HT	55	24
IIIB	Present	Asymptomatic (found on examination)	Vaginal vault + pelvic LN	HT	9	5
IIIB	Present	Vaginal bleeding	1st—vaginal vault 2nd—pelvic LN + bones	1st SE 2nd BSC	37	14

LN = lymph nodes; RT = radiotherapy; CT = chemotherapy; HT = hormonotherapy; BSC = best supportive care; SE = surgical excision.

## Data Availability

Data are unavailable due to privacy restrictions.

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
