# Peer review of "Challenging the Binary Classification of Endometrioid Endometrial Adenocarcinoma: The Evaluation of Grade 2 as an Independent Entity Based on Prognostic Characteristics and Recurrence Patterns†"

_cancers, 2025, doi:10.3390/cancers17010127_

Round 1

Reviewer 1 Report

Comments and Suggestions for Authors

 Andreas Zouridis et al. in “Challenging the binary classification of endometrioid endometrial adenocarcinoma: The evaluation of Grade 2 as an independent entity based on prognostic characteristics and recurrence patterns” show how G2 EEC is a distinct entity, thus the grading 232 system should continue to differentiate G1 EEC and G2 EEC for better prognosis interpre-233 tation. Cervical and parametrial involvement are independent risk factors for cancer spe-234 cific mortality for G2 EEC.. Finally, PIFU is expected to reduce waiting times and waiting lists due 228 to net reduction in follow-up appointments

I consider original the proposal to suggest that grading system should continue to differentiate G1 EEC and  G2 EEC for better prognosis interpretation.

The references are appropriate and recent. They support the conceptualizations present in the anticle.

The authors should better elucidate the “The switch from the traditional three tier to the binary grading system” (Line 58).

Comments on the Quality of English Language

The English could be improved to more clearly express the research.

Author Response

Comment 1: The authors should better elucidate the “The switch from the traditional three tier to the binary grading system” (Line 58).

Response 1: Thank you for your comment.  In our revised manuscript, the statement “The switch from the traditional three tier to the binary grading system was not imposed by the prognostic similarities between G1 and G2 but was a result of the diagnostic challenge of distinguishing between G1 and G2” (line 62-64)  is a conclusion based on the previous paragraph, where we explain that “However, many clinicians combine G1 and G2 tumours to a single low-grade category to increase inter- and intra- observer agreement and reproducibility of diagnosis5. That binary tumour grading system has been adopted by FIGO6 and ESMO/ESTRO/ESP7.” (lines 58-61) The relevant references 5,6 & 7 give the reader the opportunity to get more information about the rationale behind the switch from the three tier to the binary system.

Reviewer 2 Report

Comments and Suggestions for Authors

In this paper, the authors challenge the recently popular binary classification system that combines grade 1 and grade 2 tumors under the low-grade umbrella. By analyzing data from a retrospective study, the authors are able to show the differences between grade 1 and 2 groups in terms of survival and hence conclude that there is a need to differentiate grade 1 and 2 groups instead of mixing them. Some risk factors are also identified in the study. Overall, I feel that the paper is nicely written, and the message is simple, clear, and sound. My only suggestion is that the authors might consider some global tests, such as the likelihood ratio test, or even some meta-analysis approaches, like the p-value combination test, for testing a subset of predictors to increase the power of identifying the risk factors. 

Author Response

Comment 1: My only suggestion is that the authors might consider some global tests, such as the likelihood ratio test, or even some meta-analysis approaches, like the p-value combination test, for testing a subset of predictors to increase the power of identifying the risk factors. 

Response1: Thank you for your interesting point. We agree that we have failed to find independent risk factors for recurrence for G2 EEC. The reason for that is most likely our small sample and the relatively low risk of recurrence.  However, in order to implement meta-analysis approaches we should perform a systematic review of the literature first, which is beyond the scope of our paper. Moreover, most editors discourage the publication of new data and syetematic reviews on the same paper. However, a systematic review and meta-analysis would be a great idea for another paper, where our data from the present study can be included.